# Chemical Composition Profiling and Antifungal Activity of Saffron Petal Extract

**DOI:** 10.3390/molecules27248742

**Published:** 2022-12-09

**Authors:** Nadia Naim, Marie-Laure Fauconnier, Nabil Ennahli, Abdessalem Tahiri, Mohammed Baala, Ilham Madani, Said Ennahli, Rachid Lahlali

**Affiliations:** 1Department of Arboriculture-Viticulture, Ecole Nationale d’Agriculture de Meknès, Km10, Rte Haj Kaddour, BP S/40, Meknès 50001, Morocco; 2Faculty of Sciences, Moulay Ismail University, Meknes 50001, Morocco; 3Gembloux AgroBiotech, University of Liege, 5030 Gembloux, Belgium; 4Phytopathology Unit, Department of Plant Protection, Ecole Nationale d’Agriculture de Meknès, Km10, Rte Haj Kaddour, BP S/40, Menkes 50001, Morocco

**Keywords:** saffron petal extracts, antifungal effect, postharvest diseases, biochemical analysis, GC-MS, FT-IR

## Abstract

Numerous fungal plant pathogens can infect fresh fruits and vegetables during transit and storage conditions. The resulting infections were mainly controlled by synthetic fungicides, but their application has many drawbacks associated with the threatened environment and human health. Therefore, the use of natural plants with antimicrobial potential could be a promising alternative to overcome the side effects of fungicides. In this regard, this study aimed at evaluating the antifungal activity potential of saffron petal extract (SPE) against three mains important fungal pathogens: *Rhizopus stolonifer*, *Penicillium digitatum* and *Botritys cinerea,* which cause rot decay on the tomato, orange and apple fruits, respectively. In addition, the organic composition of SPE was characterized by attenuated total reflection Fourier transform infrared (ATR-FT-IR) spectroscopy and its biochemical, and gas chromatography-mass spectrometry (GC-MS) analyses were carried out. The obtained results highlighted an increased inhibition rate of the mycelial growth and spore germination of the three pathogenic fungi with increasing SPE concentrations. The mycelial growth and spore germination were completely inhibited at 10% of the SPE for *Rhizopus stolonifer* and *Penicillium digitatum* and at 5% for *B. cinerea*. Interestingly, the in vivo test showed the complete suppression of *Rhizopus* rot by the SPE at 10%, and a significant reduction of the severity of grey mold disease (37.19%) and green mold, when applied at 5 and 10%, respectively. The FT-IR spectra showed characteristic peaks and a variety of functional groups, which confirmed that SPE contains phenolic and flavonoid components. In addition, The average value of the total phenolic content, flavonoid content and half-maximal inhibitory concentration (IC_50_) were 3.09 ± 0.012 mg GAE/g DW, 0.92 ± 0.004 mg QE/g DW and 235.15 ± 2.12 µg/mL, respectively. A volatile analysis showed that the most dominant component in the saffron petal is 2(5H)-Furanone (92.10%). Taken together, it was concluded that SPE could be used as an alternative to antioxidant and antifungal compounds for the control of postharvest diseases in fruits.

## 1. Introduction

Saffron (*Crocus sativus* L.) is one of the most valuable medicinal plants worldwide. The flowers of the saffron are a combination of six petals, three stamens, and three red stigmas [1]. The dried red stigma of the saffron flowers is one of the most expensive spices in the world [2,3]. It is widely used as a spice and as a coloring and flavoring agent in the preparation of various foods, cosmetics preparation and disease treatments. The saffron is also well known for its pharmacological benefits, such as antioxidant [4], anti-inflammatory [5], antihypertensive and hypolipidemic [6], antidepressant [7] and antitumor activities [8]. Recently, a new study demonstrated its anti-inflammatory and antiviral potential against severe COVID-19 symptoms [9].

In saffron production, great amounts of floral bio-residues are generated (92.6 g per 100 g of flowers). For every kilogram of the produced spice, about 63 kg of floral bio-residues are generated (about 53 kg of petals, 9 kg of stamens, 1 kg of styles, 1500 kg of leaves,100 kg of spathes and 100 kg of corms [10]). The saffron petal, as a by-product, is available for free and produced in large amounts, compared to the saffron stigma; but in general, they are not used as a food component and are thrown away after harvesting [11] or used to feed domestic animals [12] (Moshiri et al., 2006). Diverse compounds are identified in saffron petals, such as the phenolic content and antioxidant activity [13]. Flavonols, such as kaempferol, quercetin, isorhamnetin, and anthocyanins, such as delphinidin, petunidin and malvidin, are isolated from the saffron petals [14].

Several properties of the saffron floral bio-residues have been demonstrated, such as antityrosinase [15], antidepressant [12], antinociceptive and anti-inflammatory activities [16], antifungal and cytotoxic against tumor cell lines [17], arterial pressure reducer [18] and antibacterial [19]. Therefore, saffron petals might be considered as an appropriate source for different purposes. Regarding the toxicity of saffron petals, the toxicity of the stigma is greater than the petals (the IC_50_ values of the saffron stigma and petals, in mice, were 1.6 and 6 g/kg, respectively) [20].

Fruits and vegetables are metabolically active and subjected to senescence changes that need to be controlled, to maintain their long-term quality and shelf life [21]. Generally, the postharvest decay of fruit and vegetables is caused by several plant pathogens, in particular fungi and bacteria, resulting in severe losses during packing and storing [22]. The most important fungal pathogens the cause postharvest diseases in fruits belong to the *Alternaria*, *Aspergillus*, *Botrytis*, *Fusarium*, *Geotrichum*, *Gloeosporium*, *Mucor*, *Monilinia*, *Penicillium*, and *Rhizopus* genera. These pathogens are mainly controlled using synthetic fungicides, which have several drawbacks, such as high costs, risks associated with handling, residue persistence on food, and therefore a high risk for human health and the environment [23]. As a result, consumers tend to look for residue-free products, thereby, farmers are shifting towards natural alternatives, to protect their fruit during the storing period.

Plants produce several secondary metabolites that have a biocidal action against postharvest pathogens [24]. These compounds are associated with the plant immune system and can play an important role as fungal inhibitors [25]. Numerous studies have highlighted the antimicrobial properties of natural plant extracts, basically due to their richness, with different classes of phenolic compounds [26]. In this regard, Kaveh [27] reported that the phytochemical composition of saffron petals and stigma was flavonoids, anthocyanins, alkaloids, carbohydrate glycosides, tannins, terpenes, steroids and saponins, which are useful in extending the shelf life of fresh-cut fruits, such as the watermelon [27]. Therefore, this study aims to evaluate the antifungal activity of saffron petal extracts (SPEs) in controlling postharvest diseases in fruit caused by *Rhizopus stolonifer, Penicillium digitatum* and *Botrytis cinerea.* In addition, the ATR-FTIR, GS-MS and biochemical analysis were performed to decipher the organic and chemical profiling of SPE.

## 2. Results

### 2.1. Antifungal Activity of the Saffron Petal Extract (SPE) on the Mycelial Growth

The effect of the SPE on the mycelial growth of *R. stolonifer* was revealed to be significant (*p* < 0.05) (Figure 1). The four SPE concentrations significantly reduced the mycelial growth with inhibition rates ranging from 37.62 to 100%. A complete inhibition was obtained with 10% of the SPE, which was comparable to that obtained with the fungicidal difenoconazole (1 ppm). Additionally, the IC_50_ value was determined using the linear regression equation (y = 8.007x + 19.23; R^2^ = 0.99). The estimated IC_50_ was 3.84%.

The impact of the SPE on the mycelial growth of *P. digitatum* was evaluated (Figure 1). All tested concentrations showed a significant reduction of the mycelial growth. The inhibition rate increased with the increasing SPE concentration with the percentage ranging from 37.06 to 100%. The complete inhibition was obtained with the highest concentration of the SPE (10%). This result was comparable to that obtained with the fungicidal difenoconazole (1 ppm). Furthermore, the IC_50_ value was 3.91%, according to the linear regression curve (y = 8.312x + 17.46, R^2^ = 0.95).

Regarding the effect of the SPE on the mycelial growth of *B. cinerea*, the results evidenced the same trend (Figure 1). The effect was significant with inhibition rates ranging from 25.96 to 100%. The highest inhibition rate (100%) was observed at 5% of the SPE and it was statistically comparable to that of the fungicidal difenoconazole (1 ppm). Moreover, the IC_50_ value was 1.56%, according to the linear regression equation (y = 15.04x + 26.45, R^2^ = 0.96).

### 2.2. Antifungal Activity of the Saffron Petal Extract (SPE) on the Spore Germination

The possible effect of the SPE on the spore germination of postharvest fungal pathogens was also evaluated (Figure 2). The effect was revealed as significant (*p* < 0.05). The inhibition rates of the spore germination of *R. stolonifer* ranged from 46.04 to 100%, with the highest rate observed at 10% of the SPE. The IC_50_ was 2.06%, according to the linear regression equation (y = 6.481x + 36.63; R^2^ = 0.98).

Concerning the impact of the SPE on the spore germination of *P. digitatum*, the results presented in Figure 2 denote a significant difference in the inhibition rates of the spore germination, with respect to the SPE concentrations (*p* < 0.05). The inhibition rates ranged from 48.09 to 99.47%. The spore germination was completely inhibited at the concentration of 10% SPE after 24 h of incubation. A similar result was obtained with the fungicidal difenoconazole. Furthermore, the IC_50_ was 2.07%, according to the linear regression equation (y = 6.11x + 37.28; R^2^ = 0.96).

Likewise, the effect of the SPE on the spore germination of *B. cinerea* was significant (Figure 2). The highest inhibition rate (79.70%) was obtained with 5% of the SPE and the fungicidal difenoconazole (100%). The IC_50_ was 0.66, based on the linear regression curve (y = 7.20x + 45.24; R^2^ = 0.96).

### 2.3. Effect of the Saffron Petal Extract on the Rot Decay Development

To confirm the effectiveness of the SPE in controlling the rot decay in fruit, trials were conducted and the disease severity was determined for each fungi (Figure 3). The results showed that the SPE at 2, 3 and 5% reduced the severity of *R. stolonifer* to 63.75, 62.66 and 57.96%. While, at 10% of the SPE, the disease was completely controlled (0% severity). This result was similar to that obtained with difenoconazole (1 ppm).

For green mold on the orange, the obtained results underlined a slight reduction in the disease severity, when compared with the untreated control. The highest reduction rate was registered at 10% of the SPE with 66.55%. This result was lower than that obtained with the fungicidal difenoconazole, which completely inhibited the disease development.

The disease severity, due to *B. cinerea* on the apple, was significantly reduced with reduction rates varying from 75.12 to 37.19%. The highest reduction rate was obtained with 5% of the SPE.

### 2.4. Chemical Composition

#### 2.4.1. FTIR Analysis of the Organic Composition

The ATR-FTIR spectrum was used to identify the functional groups of the active components of the SPE. The obtained ATR-FTIR spectrum of the SPE sample is shown in Figure 4. The results showed distinct peaks characteristic of the functional groups. These functional groups were identified for the saffron petals, based on the literature [28,29,30,31]. The absorption band at 3300 cm^−1^ corresponds to the stretching vibration of the O-H hydroxyl group (water or phenol and alcohol). The characteristic absorption bands at 2922 and 2850 cm^−1^ were attributed to the asymmetrical and symmetrical C-H stretch vibrations of methylene [32]. The band at 1733 cm^−1^ is due to the stretching of the carbonyl and ester groups. The band at 1607 cm^−1^ is assigned to the –C=C group and conjugated C=O group. Other characteristic vibrations of the saffron petals, attributed to the monoterpenes, are located at 1408 and 1370 cm^−1^. In addition, the spectrum showed the strong band at 1016 cm^−1^, associated with the presence of the carbohydrates group. The intensity of the bands, in ascending order, was 0.05 (3300 cm^−1^), 0.043 (1016 cm^−1^), 0.042 (2918 cm^−1^), 0.035 (2850 cm^−1^), 0.023 (1607 and 1370 cm^−1^), 0.22 (1409 cm^−1^), 0.021 (1640 cm^−1^) and 0.018 (1733 cm^−1^). The band at 3300 cm^−1^ was mainly associated with the effect of the antioxidant and antifungal, while other bands were related to the lipid acyl chains, carbonyl ester group, phenolic, aromatic groups and the presence of cell wall components (Table 1).

#### 2.4.2. Total Phenolic, Flavonoid Contents and the DPPH Radical Scavenging Activity

The total phenolic, flavonoid contents and the DPPH radical scavenging activity in the saffron petal extract was quantified (Table 2). The average value of the total phenolic and flavonoid contents and the half-maximal inhibitory concentration (IC_50_) were 3.09 ± 0.012 mg GAE/g DW, 0.92 ± 0.004 mg QE/g DW and 235.15 ± 2.12 µg/mL, respectively.

#### 2.4.3. Volatile Composition: GC-MS

In order to identify the volatile compounds in the SPE, GC-MS chromatography was performed. The obtained results are listed in Table 3 and show that the most dominant component in the volatile fraction is 2(5H)-Furanone (92.10%). The safranal (3.56%) and limonene (1.48%) were also identified at low levels. In addition, other compounds were also found in trace amounts (Table 3).

## 3. Discussion

The control of postharvest pathogens of fruits and vegetables is mainly achieved by applying fungicides in pre/postharvest periods, which might have several disadvantages, such as the persistence of residues on fruits, which is a high risk for human health and the environment, the high cost and the appearance of fungicide resistant strains in the pathogen population [23]. In this study, the effect of the saffron petal extract was investigated for the control of postharvest diseases in fruits as a potential alternative strategy to replace the use of chemicals. It was known that plants are capable of producing a range variety of secondary metabolites that have antifungal activities against major postharvest pathogens [24]. These compounds are associated with the immune plant system and can play a major role as fungal inhibitors [25]. The saffron petal is the main by-product of *Crocus sativus,* that is produced in large quantities and is known for its several properties, particularly its antimicrobial potential, which could be a good alternative for controlling postharvest fungal infections. Furthermore, the antifungal effect of the saffron petal extract was evaluated against three most important fungal pathogens causing postharvest damage to the tomato, orange and apple.

To the best of our knowledge, there is currently no report on the ability of the saffron petal extract to suppress postharvest diseases. The promising findings from this study showed a great inhibitory effect of the petals of the saffron, suggested that the saffron petal extract might have metabolites with a higher antifungal activity against *R. stolonifer* on the tomato and a moderate significant reduction of grey mold on the apple and a slight inhibition of green mold on the orange. These findings evidenced that the saffron petal extract has antifungal [17] and antimicrobial effects [27]. In this regard, our in vitro trials showed that the mycelial growth and spore germination of *R. stolonifer* and *P. digitatum* were completely inhibited at 10% of the petal extract. Interestingly, the growth of *B. cinerea* was inhibited at 5% of the SPE. Previous studies demonstrated the ability of plant extracts to reduce postharvest fungal diseases [24]. Jasso de Rodríguez et al. [33] reported that the mycelial growth of *R. stolonifer* was completely stopped at 3 g/L of the *Flourensia* spp extract. A similar result was obtained in another study in which the complete inhibition of the spore germination of *P. digitatum* and *B. cinerea* was observed when the pomegranate peel aqueous extract was used at a concentration of 12 g/L after 20 h of incubation [34]. Furthermore, Gholamnezhad [35] highlighted the in vitro efficacy of seven plant extracts (neem, fennel, lavender, thyme, pennyroyal, salvia and asafetida) to reduce the mycelial growth of *B. cinerea* [35]. Interestingly, our results showed that the inhibition rate of the mycelial growth and spore germination increased with the increasing SPE concentrations, regardless of the fungal species. These results were confirmed by the in vivo trials in which the disease severity was reduced with the increasing SPE concentration. López-Anchondo et al. [36] found that the antifungal effect is proportional to the increase in the concentration of the extract and the *Prosopis glandulosa* extract had an antifungal index of 55% for *R. stolonifer*. Interestingly, the SPE at 10%, completely suppressed the disease in artificially injured and inoculated fruit by *R. stolonifer* (0% severity) and result was similar to the fungicidal difenoconazole (1 µg/mL). The SPE had no phytotoxic effect on the tissues of the tomato fruit. This finding is very interesting, compared to other studies conducted using other plant extracts. Lopez-Anchondo et al., 2020 found that the application of the *P. glandulosa* extract displayed less efficiency in controlling the *Rhizopus* species [36]. Moreover, our results demonstrated that applying the SPE at 5%, significantly reduce the disease severity of grey mold (37.19%) on the apple, compared to the untreated control in which the disease severity reached 100%. In a similar study, Gholamnezhad [35] found that the *Azadirachta indica* methanolic and aqueous extracts, applied at 25%, significantly reduced the disease on the wounded area by 52 and 89%, respectively, compared to the control [35]. Surprisingly, a slight reduction of the disease severity, due *P. digitatum*, was obtained at 10% of the SPE.

In order to understand the mechanisms by which the SPE control postharvest fungal pathogens, a series of chemical analyses were undertaken in this study. Among them, the FT-IR spectroscopy, which is an effective tool to detect different chemical components in food products [37,38]. The obtained results revealed that the SPE has potent antifungal properties, which may be attributed to the presence of many chemical components, including phenols-alcohols (O-H), aromatic group and monoterpene composites (C-H), which can be the main chemical compounds that affect the biological activity of saffron petals. This result was confirmed by the phytochemical quality of the SPE, which highlighted the implication of the phenolic and flavonoid components contained in the SPE in its microbial activity. The phenolic contents are highlighted as very powerful antimicrobial agents that exert a direct effect by neutralizing the microbial systems and damaging the hyphae [39]. Anthocyanins are responsible for the attractive color of the saffron petals, among which delphinidin, petunidin and malvidin glycosides represent 30% of the total content of the phenolic compounds in the petals [1]. Likewise, De Leon-Zapata et al. [40] reported that the fungal inhibition is correlated to the concentration of the bioactive compounds of the tarbush leaf extract, and especially to the gallic acid and flavonoids [38]. They found that, in vitro, the highest inhibition mycelial growth of *R. stolonifer* was 67.40% at 4 g/L.

Flavonoids are important constituents of plants because of the scavenging ability conferred by their hydroxyl groups. The flavonoids may contribute directly to anti-oxidative and antimicrobial actions [41]. Indeed, Termentzi and Kokkalou [42] found that the saffron petal is a good potential source of quercetin, kaempferol and naringenin, which are relatively highly resistant flavonols to thermal degradation [42]. In addition, saffron petals have been shown to have a higher antioxidant activity and their beneficial effects, derived from phenolic compounds, are usually attributed to their antioxidant activity [43,44]. These results are consistent with several previous studies [13,18,45]. In addition, several volatile compounds were found in the SPE, of which furanone is the most predominant. Interestingly, a previous study evidenced the biological activity of furanone against some germs [46]. Similarly, several studies reported that a large number of halogenated furanones and related synthetic analogues, were later discovered to inhibit the biofilm formation in a variety of pathogens [47,48,49]. Therefore, the richness of the SPE with furanone might explain its higher antifungal activity in this study. Ultimately, compounds of natural origin with an antifungal activity are present in several plants [50]. The plant activity is determined by the plant genotype and depends on their chemical composition, which is influenced by several factors, including environmental conditions and geographical location [51]. The mechanisms of action by which plant extracts suppress the growth of postharvest fungal pathogens, are multiple and include the disturbances in the cell membrane function, the disruption of the energy activity and damage of the cytoplasmic membrane [52]. In addition, a previous study conducted by Ma et al. [53] also focused on the control of *B. cinerea* by using honokiol, a poly-phenolic compound obtained from *Magnolia officinalis*. It was found that honokiol significantly inhibited the mycelial growth and reduced the virulence of *B. cinerea*. It was demonstrated also that honokiol altered the mitochondrial membrane potential with the accumulation of the reactive oxygen species. Moreover, some key genes involved in the fungal pathogenicity have their expression down-regulated. In a recent study, Mastino et al. [54] underlined that the phenolic compounds represent a rich source of protectants and biocides, which can be used as alternative strategies for the control of postharvest diseases in fruits [54]. Rubio-Moraga et al. [55] pointed out that saponins and phenolic compounds could be responsible for the fungicidal activity detected in internal parts of the corms, against five fungi [55]. According to Zhang et al. [56], The effectiveness of the antifungal activity of the plant extracts is correlated with the extraction process, particularly the interaction between the solvent and the raw material, which allows its dissolution and separation from the solid matrix, depending on the solvent/solid ratio, particle size, temperature and the timing of the extraction [56]. Furthermore, the use of innovative processing techniques, such as microwave-assisted extraction, ultrasound-assisted extraction and ohmic heating assisted extraction, was proved to have a substantial effect on the antifungal activity of the jackfruit extract against fungal pathogens *Colletotrichum gloeosporioides* and *Penicillium italicum* [57]. Thereby, natural molecules generated by the plant present many advantages for the consumer because it protects against toxic substances produced, either by postharvest fungal pathogens or biocontrol antagonists, and therefore they present an additive food for human health [58].

## 4. Materials and Methods

### 4.1. Collection and Preparation of the Saffron Petal Aqueous Extract

The dry saffron petals were collected in November 2021 after pruning the harvest from a saffron farm in Serghina/Boulmane. The petals are placed in an oven at 37 °C to drive out the humidity. The saffron petals were crushed using an automated grinder and then stored until use. For the in vitro test, different concentrations (0.5, 1, 2, 3, 5, 7, and 10%) of the appropriate amount of the powder were added to 100 mL of distilled water, to achieve the desired concentration. The suspensions obtained were brought to a boil, filtered and mixed with potato dextrose agar (PDA). Then, they were autoclaved for 20 min at 121 °C, before being distributed into 9 cm diameter Petri dishes [52,53,54].

### 4.2. Fungal Pathogens

The fungal pathogens used to assess the efficacy of the aqueous extract of the saffron petals were *P. digitatum, B. cinerea* and *R. stolonifer*, which were provided by the laboratory of the Department of Plant Protection and Environment, Phytopathology Unit, Ecole Nationale d’Agriculture, Meknes, Morocco. Prior to the testing, the sub-fungal isolates were subcultured on a potato dextrose agar medium (PDA). The spore suspension was obtained by scraping the fungus and mixing it with 20 mL of sterile distilled water (SDW). The resulting liquid was filtered through a sterile filter, to remove the hyphal fragments and medium debris after centrifugation.

### 4.3. Fruit Preparation

Navel oranges (Lane late), Golden delicious apples and tomatoes were used to study the in vivo effects of the saffron petal extract on green rot caused by *P. digitatum*, and gray rot caused by *B. cinerea* and *Rhizopus* rot, respectively. All fruits were bought from the local market in the town of Meknes. They were washed, disinfected with 2% (*v*/*v*) sodium hypochlorite, rinsed three times in sterile distilled water and then air dried for 1 h at room temperature under a laminar flow cabinet. Once dried, two artificial wounds (4 mm in diameter and 3 mm in depth) were performed on both equatorial sides of each fruit with a sterile cork-borer [59].

### 4.4. Antifungal Activity of the Saffron Petal Extract (SPE) on the Mycelial Growth

The agar dilution method was used to determine the ability of the saffron petal extract to inhibit the mycelial growth of *P. digitatum, B. cinerea* and *R. stolonifer*. The saffron petal extract was tested at different concentrations: 0.5, 1, 2, and 5% for *B. cinerea*, 2, 5, 7 and 10% for *P. digitatum*, and 2, 3, 5 and 10%, and for *R. stolonifer*. Each Petri plate was aseptically inoculated with each pathogen, using a 5 mm mycelium taken from a 7-day-old colony. The Petri dishes were sealed with parafilm and incubated at 25 °C for 5 days. The pathogens cultured in PDA without the extract, served as a control. Each treatment was repeated three times and the antifungal activity that was expressed as the inhibition rate was compared to the control and was calculated after 5 days of incubation, according to the following formula:Mycelial growth inhibition rate (MGI) = [(colony diameter on control treatment − colony diameter on SPE treatment)/colony diameter on control treatment] × 100

### 4.5. Effect of the Saffron Petal Extract on the Spore Germination

The method used to study the effect of the saffron petal extract on the spore germination of each fungus consisted of mixing the spore suspension (1 × 10^4^ spores/mL) with each aqueous extract concentration at an equal volume (1 v/1 v) as following: 0.5, 1, 2, and 5% for *B. cinerea*, 2, 5, 10, and 12% for *P. digitatum*, and 2, 3, 5, and 10% for *R. stolonifer*. The control consisted of using the same amount of spore suspension without the plant extract. The mixtures were incubated at 24 °C in sterile micro-centrifuge tubes. The spore germination was examined under a light microscope after 24 h. At least 100 spores were observed for each replicate at 40× magnification. The inhibition rate of the spore germination was determined, according to the following formula:GI (%) = [(Gc − Gt)/Gc)] × 100
where, Gc and Gt represent the mean number of the germinated spores in the control and treated tubes, respectively.

### 4.6. Effect of the Saffron Petal Extract on the Rot Decay Development

The in vivo test consisted of treating the disinfected and wounded fruits with 50 µL of the plant extract at different concentrations. Following 2 h of incubation at room temperature, under a laminar flow cabinet, each wound was inoculated with 20 µL of the spore suspension concentrated at 1 × 10^4^ spores/mL. The fruits treated with sterile distilled water (SDW) and difenoconazole fungicide (15 µL/10 mL) were served as controls. The treated fruits were weighed and placed in plastic bags and incubated in darkness at 24 °C with 95% relative humidity (RH). Two experiments were performed over time with three replicates for each concentration. Then, 7 days later, the disease severity (%) was calculated for all treatments (plant extract, water control and fungicide control), according to the following formula:Disease Severity (%) = [(average lesion diameter of treatment/average lesion diameter of control)] × 100

### 4.7. Chemical Composition Analysis of the Saffron Petal

#### 4.7.1. FTIR Analysis

A ground and homogenized saffron petal sample was scanned in the wavelength range of 4000–400 cm^−1^ with a spectral resolution of 4 cm^−1^ using the FTIR spectrometer (PerkinElmer, Waltham, MA, USA) and the characteristic peaks and their functional groups were detected. The FTIR peak values were recorded. The analysis was repeated three times and the averaged spectrum was used.

#### 4.7.2. Determination of the Total Phenolic and Flavonoid Contents and the DPPH Radical Scavenging Activity

The total phenolic and flavonoid contents were determined for the SPE concentration 10 mg/mL. The extraction was based on a method previously described by Ghanbari et al. [60] using methanolic solutions of the extract.

The total phenolic content (TPC) of the saffron petal extract was determined by a colorimetric method, based on the procedure described by Ghanbari et al. [60]. Briefly, 0.5 mL of extract was added to 2.5 mL of Folin–Ciocalteu (FC) reagent (1:10) and incubated for 5 min at room temperature. Then, 2 mL of 7.5% sodium carbonate solution was added. Once shaken, the mixture was incubated in a hot water bath at 45 °C for 15 min. Finally, the absorbance was recorded at 765 nm. The results were expressed as mg of the gallic acid equivalent (GAE/g sample dry weight (DW)).

The total flavonoid content was measured by the aluminium chloride method using quercetin as a standard and described by Ghanbari et al. [60]: 0.3 mL of 5% NaNO2 solution was added to 0.5 mL of the methanolic extract. The mixture was incubated in the dark at room temperature for 6 min. Thereafter, 0.6 mL of 10% AlCl_3_ was added and incubated for 5 min. Finally, 3 mL of NaOH 1M was added, and the final volume was adjusted to 10 mL with distilled water. The absorbance was read at 510 nm after 15 min incubation. The total flavonoid content values were expressed as mg of the quercetin equivalent (QE) per g DW.

The methanolic DPPH solution 0.5 mM (1.5 mL) was added to 0.75 mL of prepared 50, 100 and 300 µg/mL extract concentrations [60,61]. Then, 20 min later, the absorbance was determined at 517 nm with 80% methanol as blank. The same concentrations of ascorbic acid were used as a positive control. The percentage of the inhibition was determined, according to the following formula: Inhibition rate (%) = ((A control − A sample)/A control) × 100, where A sample is the absorbance value of the sample and A control is the absorbance of the control. Following the calculation of the percentage of the inhibition, a linear regression model was established, based on the concentration and percentage of the inhibition.

#### 4.7.3. GC-MS Analysis

The volatile components analysis of the saffron was carried out using gas chromatography-mass spectrometry (GC-MS) equipped with an Agilent 7890A system (A.01.01, Wilmington, DE, USA) and a mass selective detector 5975 Network MSD and coupled to a MPS automatic sampling system, as described previously by Naim et al. [62]. The chromatographic separation was performed on a HP-5MS capillary column (30 m × 0.25 mm, film thickness 0.17 mm), and the following temperature program was used: 60 °C held for 3 min, then increased to 210 °C at a rate of 4 °C/min, then held at 210 °C for 15 min, then increased to 300 °C at a rate of 10 °C/min, and finally held at 300 °C for 15 min. Helium was used as the carrier gas at a constant flow of 1 mL/min. For the quantification, the results are presented as a percentage of the peak area, considering a response factor of the fiber. Mass Hunter Version B.06.00 (Agilent Technologies) was used for the data acquisition and processing. The identification of the components was based on the comparison of the obtained mass spectrum with those from the commercial databases (NIST17 and Wiley) and by comparison with the retention index (RI) of each peak from the literature (Pherobase). The experimental retention index (RI) of the compounds were calculated following the injection of a mixture of n-alkanes C8-C20 (Sigma Aldrich, Darmstadt, Germany).

### 4.8. Statistical Analysis

The statistical analysis was performed using SPSS V25 software (version 25, IBM SPSS Statistics 20, New York, NY, USA) and the datasets were expressed as the mean ± standard deviation. Duncan’s multiple analysis was used for the means separation at a significance level of *p* ≤ 0.05. The linear regression equation of the mycelial growth and the spore germination inhibition rates versus the logarithmic of the SPE concentrations were performed to calculate the half-maximal effective concentration (IC_50_).

## 5. Conclusions

In this study, the chemical composition analysis of the saffron petal extract was carried out and its antifungal activity was investigated. In light of these findings, it was concluded that the SPE could be used to reduce postharvest fruit infections caused by fungal pathogens, such as *R. stolonifer*, *B. cinerea* and *P. digitatum*. The antifungal activity of the SPE might be explained by its antioxidant power and its richness in phenolic and flavonoid contents. In addition, the use of the SPE does not present any risks to both the user and consumer. Therefore, this study has shed light on new opportunities of using the SPE to control postharvest fruit infections and could be used as an alternative to chemical products. However, further investigations are needed to assess the effectiveness of the SPE to control fungal plant diseases in large-scale trials.

## Figures and Tables

**Figure 1 molecules-27-08742-f001:**
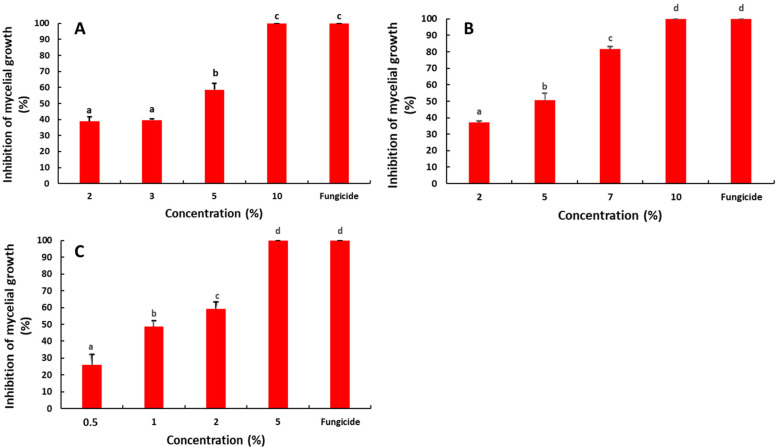
Effect of the saffron petal extract on the mycelial growth of postharvest fungal pathogens *R. stolonifera* (**A**), *P. digitatum* (**B**) and *B. cinerea* (**C**). The different letters (a–d) represent the statistically significant differences between the concentrations, according to Duncan‘s test (*p* < 0.05).

**Figure 2 molecules-27-08742-f002:**
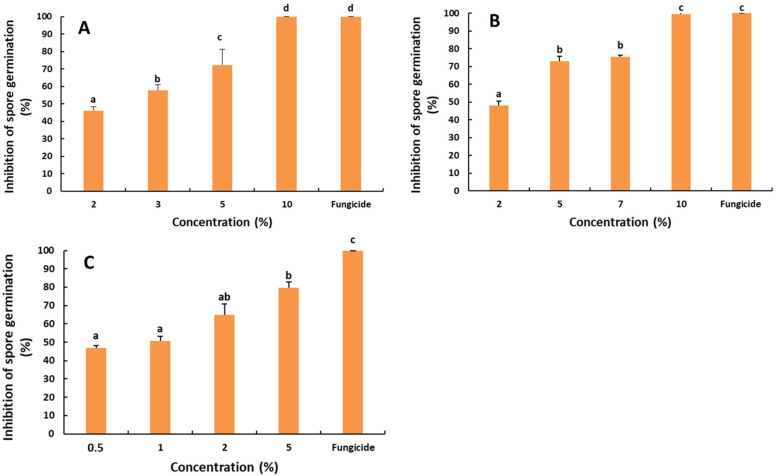
Effect of the saffron petal extract on the spore germination of postharvest fungal pathogens. *R. stolonifera* (**A**), *P. digitatum* (**B**) and *B. cinerea* (**C**). The different letters (a–d) represent the statistically significant differences between the concentrations, according to Duncan‘s test (*p* < 0.05).

**Figure 3 molecules-27-08742-f003:**
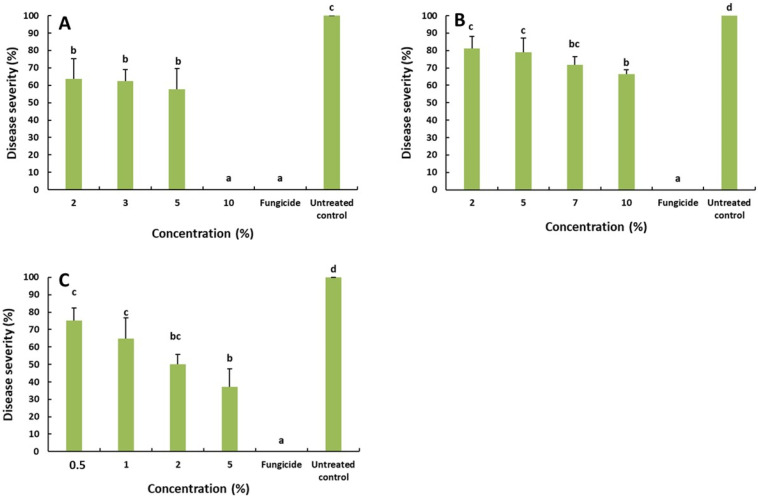
Effect of the saffron petal extract on the development of rot decay caused by *R. stolonifer* (**A**)on the tomato, *P. digitatum* (**B**) on the orange and *B. cinerea* (**C**) on the apple fruit. The different letters (a–d) represent the statistically significant differences between concentrations, according to Duncan‘s test (*p* < 0.05).

**Figure 4 molecules-27-08742-f004:**
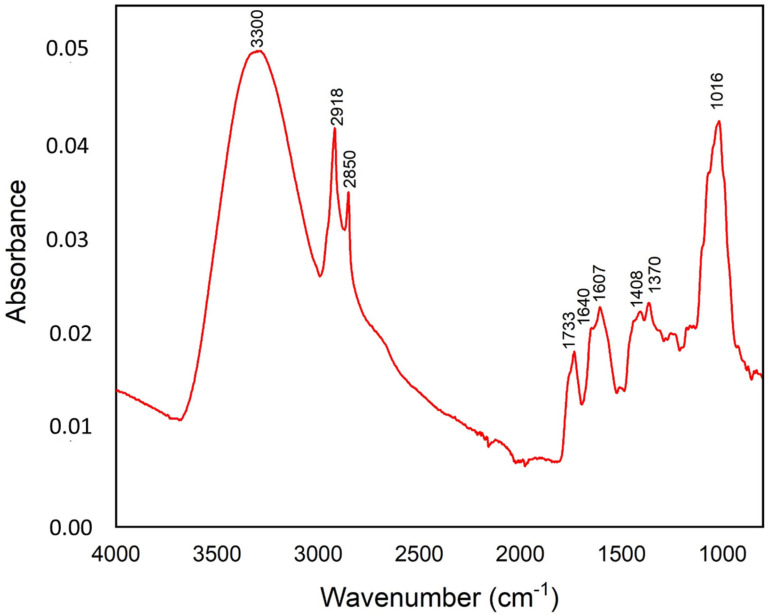
Averaged ATR-FTIR spectrum of the saffron petal powder in the mid-infrared region (4000–800 cm^−1^). This IR spectrum is an average of three replicates, each one corresponding to the accumulation of 128 scans.

**Table 1 molecules-27-08742-t001:** Assignment (tentative) of the infrared bands observed in the saffron petal powder, based on the literature [31,32].

Wavenumber (cm^−1^)	Group	Characteristics
3300	O-H str. of the hydroxyl group	Hydroxyl of the phenolic compounds
2918	C-H str. (asym) of CH2	Aliphatic C-H from the lipid acyl chains
2850	C-H str. (sym) of CH2 from	Aliphatic C-H from the lipid acyl chains
1733	C-O stretching vibration	Carbonyl ester group (lipid)
1640	C=O and C=C stretching vibratons of cis-alkene	Carboxylic groups, hemicellulose or amide groups in proteins
1607	–C=C group of alkenes and conjugated C=O group	Aromatic group, phenolic ring, pectin ester group
1408	C-H stretching	Aromatic group
1370	CH_2_ scissors vibration	Xyloglucan, cellulose
1016	C-O stretchings	Pectins

**Table 2 molecules-27-08742-t002:** Average levels of the total phenolic content, total flavonoid content and the DPPH radical scavenging activity.

**SPE**	**Total Phenolic (mg GAE/g DW)**	**TFC (mg QE/g DW)**	**DPPH (IC_50_) (µg/mL)**
	3.09 ± 0.012	0.92 ± 0.004	235.15 ± 2.12

Values are the means of three independent replicates.

**Table 3 molecules-27-08742-t003:** List of identified volatile compounds in the saffron petal by GC-MS.

Compound	Cas Number	RI Lit	RI Calculated	% Peak Area
2(5H)-Furanone	497-23-4	951	920	92.10
Limonene	138-86-3	1029.5	1033	1.48
Phenylethyl alcohol	60-12-8	1114.9	1120	0.70
3,5,5-trimethyl-3-cyclohexen-1-one	471-01-2	1429	1128	0.52
2,6,6-trimethyl-2-cyclohexene-1,4-dione	1125-21-9	1142	1150	0.51
Safranal	116-26-7	1201	1204	3.56
Carvone	99-49-0	1242	1227	0.53
Thymol	89-83-8	1290.1	1293	0.61

RI lit: The RI theoretical value was found in Pherobase in the same column.

## Data Availability

Data will be made available upon request.

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
