# Peer review of "Chemical Composition Profiling and Antifungal Activity of Saffron Petal Extract"

_molecules, 2022, doi:10.3390/molecules27248742_

Round 1

Reviewer 1 Report

This article presents an evaluation of the antifungal activity, along with ATR-FTIR, GS-MS and biochemical analyses of the saffron petals extract.

The study is well-researched, and the results are well-presented. This article presents interesting findings about saffron plant extract and its potential to reduce post-harvest fruit infections. I suggest the authors to check the spelling throughout the text and make the corrections as I suggested below:   

Line 54 – reference [12] appears twice.

Line 84 – the formulation “this study aimed to evaluating the antifungal activity…” is not correct. Please correct it.

The abbreviations ATR-FTIR and GS-MS should be explained where they were first introduced.

Line 90 - R. stolonifer should be italic.0

 Please check the spelling and the font style throughout the whole text for all pathogen's names.

Line 124 – misspelled “concentartions”

 Line 165 – 168. I think it is an overstatement to associate the antifungal effect of a compound with one band. Please reformulate or explain in more detail and support this affirmation with references.

Line 216 – “in vitro” should also be italic

Line 228 - “in vivo” should also be italic

Author Response

This article presents an evaluation of the antifungal activity, along with ATR-FTIR, GS-MS and biochemical analyses of the saffron petals extract.

The study is well-researched, and the results are well-presented. This article presents interesting findings about saffron plant extract and its potential to reduce post-harvest fruit infections. I suggest the authors to check the spelling throughout the text and make the corrections as I suggested below:   

Line 54 – reference [12] appears twice.

Answer: revised as suggested

Line 84 – the formulation “this study aimed to evaluating the antifungal activity…” is not correct. Please correct it.

Answer: The formulation was corrected as suggested (please see the revised version of MS).

The abbreviations ATR-FTIR and GS-MS should be explained where they were first introduced.

Answer: We agree with the reviewer’s comment, the full names of abbreviated ATR-FTIR and GC-MS were added as suggested (please see the revised version of MS).

.

Line 90 - R. stolonifer should be italic.

 Answer: corrected

 Please check the spelling and the font style throughout the whole text for all pathogen's names.

Answer: The authors revised the whole manuscript, and corrected any spotted typos in the text.

Line 124 – misspelled “concentartions”

Answer: Corrected

Line 165 – 168. I think it is an overstatement to associate the antifungal effect of a compound with one band. Please reformulate or explain in more detail and support this affirmation with references.

Answer: At 3300cm-1, the absorption band corresponds to the stretching vibration of the O‒H hydroxyl group (Hydroxyl of phenolic compounds) and the antifungal activity, which may be attributed to the presence of many chemical components including phenol-alcohol (O‒H) and monoterpene composites, which can be the main chemical components that affect the biological activity of saffron petals.

Line 216 – “in vitro” should also be italic

Answer: Corrected

Line 228 - “in vivo” should also be italic

Answer: Corrected

Reviewer 2 Report

authors selected very important and interesting topic of research

authors pointed out fungal growth inhibition in extracts treated samples, but moisture contents and humidity concern is there, which can promote bacteria growth, also fungal infections sometimes are more dangerous as far as their secondary metabolites or fungal toxins are concern, toxins can survive even when organism itself dies. this point still remains unclear. also need to rewrite conclusion section in more elaborated way so that main idea and focus of work can be reflected.

Author Response

authors selected very important and interesting topic of research

authors pointed out fungal growth inhibition in extracts treated samples, but moisture contents and humidity concern is there, which can promote bacteria growth, also fungal infections sometimes are more dangerous as far as their secondary metabolites or fungal toxins are concern, toxins can survive even when organism itself dies. this point still remains unclear. also need to rewrite conclusion section in more elaborated way so that main idea and focus of work can be reflected.

Samples were washed, disinfected with 2% (v/v) sodium hypochlorite, rinsed three times in sterile distilled water and then air dried for 1 h at room temperature under a laminar flow cabinet. Therefore, he risk of bacterial development is very low, especially since it has been proven by numerous studies that saffron extract has strong antibacterial activity against several strains of bacteria. The antibacterial activity of the saffron petals extract is also the subject of our study.

Answer: We thank the reviewer for his valuable comments; a statement was added to conclusion part to explain objectives before highlighting the outcomes!

Reviewer 3 Report

The article describes the Chemical composition profiling and Antifungal activity of saffron petals extract. Some observations:

I suggest improving the resolution of the figures in the article.

Several formatting errors in the text regarding the in vivo terms, LD50 AND IC50 will need to be corrected.

  In the characterization, de-replicate the extract by LC/MS or obtain a profile of the extract through hydrogen and carbon NMR. These data would enrich the part of the extract's characterization.

After all the corrections, the article can be accepted for publication because it brings excellent contribution showing a possible option of an extract with antifungal activity.

Although the work presents little originality in characterizing the saffron petals extract. The extract presents a high antifungal potential. However, more experiments for better characterization of the saffron petals extract are needed (for example, a dereplication by LC/MS).

Author Response

The article describes the Chemical composition profiling and Antifungal activity of saffron petals extract. Some observations:

I suggest improving the resolution of the figures in the article.

Answer: Figures are very clear to readers; we have checked it as suggested!

Several formatting errors in the text regarding the in vivo terms, LD50 AND IC50 will need to be corrected.

Answer: Corrections have been made in the revised version of the MS as per suggested.

  In the characterization, de-replicate the extract by LC/MS or obtain a profile of the extract through hydrogen and carbon NMR. These data would enrich the part of the extract's characterization.

After all the corrections, the article can be accepted for publication because it brings excellent contribution showing a possible option of an extract with antifungal activity.

Although the work presents little originality in characterizing the saffron petals extract. The extract presents a high antifungal potential. However, more experiments for better characterization of the saffron petals extract are needed (for example, a dereplication by LC/MS).

Answer: The characterization of the saffron petals extract by LC/MS is planned. We did not do it in this study due to machine problems and lack of some supplies. We'll plan to do this near future when all of these problems will be solved!

Round 2

Reviewer 2 Report

some of the concerns raised in review still need consideration and inclusion in manuscript. pointwise answer to every query is needed

Reviewer 3 Report

The authors corrected the errors in the article and answered my questions convincingly.  The article can be accepted in its present form